# Solving Offline Reinforcement Learning with Decision Tree Regression

**Prajwal Koirala**
Iowa State University
`prajwal@iastate.edu`

**Cody Fleming**
Iowa State University
`flemingc@iastate.edu`

**Abstract:** This study presents a novel approach to addressing offline reinforcement learning (RL) problems by reframing them as regression tasks that can be effectively solved using Decision Trees. Mainly, we introduce two distinct frameworks: return-conditioned and return-weighted decision tree policies (RCDTP and RWDTP), both of which achieve notable speed in agent training as well as inference, with training typically lasting less than a few minutes. Despite the simplification inherent in this reformulated approach to offline RL, our agents demonstrate performance that is at least on par with the established methods. We evaluate our methods on D4RL datasets for locomotion and manipulation, as well as other robotic tasks involving wheeled and flying robots. Additionally, we assess performance in delayed/sparse reward scenarios and highlight the explainability of these policies through action distribution and feature importance.

**Keywords:** Offline Reinforcement Learning, Decision Trees

## 1 Introduction

There have been many attempts to transcribe a reinforcement learning (RL) problem into a supervised learning (SL) problem [1, 2, 3, 4, 5]. The motivation behind this transformation primarily stems from two key factors. First, by treating RL as a SL task, the training process tends to show improved stability due to reduced susceptibility to the challenges associated with non-stationary targets. This facilitates smoother convergence. Second, the use of a true supervised learning objective enhances data efficiency, preventing it from becoming 'stale' after a single learning step. This enables agents to learn more quickly and effectively from their experiences.

However, the distinction between these two learning paradigms (SL and RL) lies in the nature of the feedback signals they utilize. In supervised learning, the error signal, typically represented as loss or cost, is generated from labeled data and minimized through gradient descent algorithms. Conversely, reinforcement learning relies on an evaluation signal provided by the environment, such as reward or return, which require special treatment depending on the setting. For example, Deep Q Networks try to estimate the Q-value of the state-action pair based on the evaluation signal, and Policy Gradient Methods adjust the action probabilities to optimize the policy accordingly [6, 7, 8]. Contemporary state-of-the-art reinforcement learning approaches often seek to encapsulate this specialized treatment through sophisticated loss functions that involve the evaluation signal term, enabling minimization akin to error signals in supervised learning paradigms.

The traditional online learning paradigm of RL involves iteratively collecting experience by interacting with the environment and using that experience to improve the policy [9]. However, in recent years, offline reinforcement learning has emerged as a promising approach in robot learning, particularly due to its utilization of previously collected data without the need for continuous online data collection. This circumvents the impracticality and expense associated with continuous data gathering in many settings. By leveraging stationary historical data, offline RL enables agents to learn without direct interaction with the environment during training, thereby transforming large

8th Conference on Robot Learning (CoRL 2024), Munich, Germany.

datasets into powerful decision-making engines. While the static nature of the dataset allows of-fline RL to be treated as a supervised learning problem—an apparently advantageous prospect—it also introduces the challenge of covariate shift, the disparity between the data distribution and the learned policy distribution, which necessitates a more careful handling [10]. In contrast to offline RL, imitation learning algorithms like Behavior Cloning (BC) and its variants treat the static dataset as labeled data (states as features and actions as labels), often disregarding the evaluation signal [11, 12, 13, 14]. Although Kumar et al. [15], in their comparison of Conservative Q-Learning (CQL [16]) with Behavior Cloning, highlight the capability of offline RL algorithms to derive effective policies even from suboptimal data, showcasing superiority over BC algorithms in conditions with sparse rewards or noisy datasets, in many cases Behavior Cloning (BC) outperforms offline RL algorithms. In addition, BCs offer faster training, minimalistic loss function with no requirement of 'special treatment', and are less sensitive to hyperparameter tuning compared to CQL [17].

In this context, we propose transforming a reinforcement learning task in an offline setting into a simple regression objective, allowing us to leverage decision trees, which have shown strong performance in regression and classification tasks, often providing a viable alternative to neural networks. By employing an 'extreme' gradient boosting algorithm for regression over actions space, we train the agent's policy as an ensemble of weak policies, achieving rapid training times of mere minutes, if not seconds. Besides replacing neural networks as the default function approximators in RL, our paper also introduces two offline RL frameworks: **Return Conditioned Decision Tree Policy (RCDTP)** and **Return Weighted Decision Tree Policy (RWDTP)**. These methods embody **simplified modeling strategies**, **expedited training methodologies**, **faster inference mechanisms**, and require **minimal hyperparameter tuning**, while also offering **explainable policies**. We conduct comprehensive comparisons with prior methods, discuss the use cases of our proposed methods, and also assess our model's performance in scenarios with delayed and sparse rewards using D4RL benchmark tasks. Finally, we explore the applicability of our methods in robot learning and real-time control of dynamical systems, highlighting their potential in practical settings.

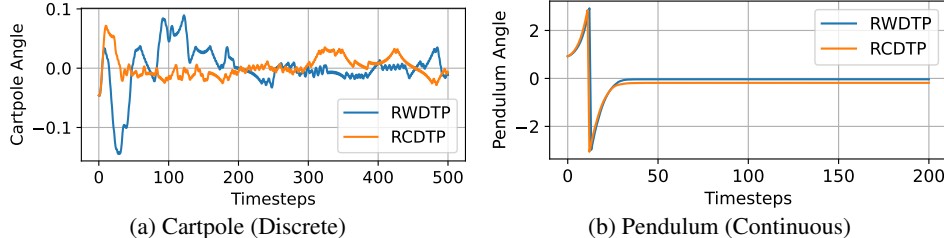

|                        |                          |
| :--------------------: | :----------------------: |
| (a) Cartpole (Discrete) | (b) Pendulum (Continuous) |

Figure 1: Decision Tree Policies in Classical Control Environments with Different Action Spaces. Both methods achieve expert-level returns in both environments using medium-level demonstration datasets from d3rlpy [18], with all trainings completed within a second.

## 2 Preliminaries and Related Works

### 2.1 Offline Reinforcement Learning

Reinforcement learning (RL) is a mathematical framework for learning-based control, where an agent learns to optimize user-specified reward functions by acquiring near-optimal behavioral skills, represented by policies. This process typically involves iteratively collecting experience by interacting with the environment and using that experience to improve the policy.

*Offline RL* focuses on learning policies from a static dataset without any active data collection. Instead of obtaining data via environment interactions like in online RL, the agent only has access to a fixed, limited dataset consisting of trajectory rollouts from an arbitrary behavior policy (or policies). The inability of the agent to further explore the environment and collect additional feedback makes it a challenging form of learning-based decision making and control.

The environment in a sequential decision-making setting like RL is typically defined as a Markov Decision Process (MDP), represented as the tuple $(\mathcal{S}, \mathcal{A}, \mathcal{P}, \mathcal{R})$, where $\mathcal{S}$ is the state space, $\mathcal{A}$ is the action space, $\mathcal{P}$ is the state transition probability function and $\mathcal{R}$ is the reward function. For some $s, s' \in \mathcal{S}$ and $a \in \mathcal{A}$, $\mathcal{P}(s'|s, a) : \mathcal{S} \times \mathcal{S} \times \mathcal{A} \rightarrow [0, 1]$ and $\mathcal{R}(s, a, s') : \mathcal{S} \times \mathcal{A} \times \mathcal{S} \rightarrow \mathbb{R}$.

A trajectory of time-length $T$ is made up of a sequence of states, actions, and rewards ($\tau = < s_t, a_t, r_t >_{t=0}^{T}$). The reward at any timestep is usually the function of the state, action and the next state. The return at timestep $t$, $R_t$, is the discounted sum of future rewards with discount factor $\gamma$: $R_t(< r_t >, \gamma) = \sum_{k=t}^{T} \gamma^{k-t} r_k$. The return-to-go (RTG) at a timestep is defined as the undiscounted sum of future rewards from that timestep (RTG= $R_t(< r_t >, \gamma = 1)$). The goal in RL (including offline RL) is to learn a policy that maximizes the expected return in an MDP.

## 2.2 Decision Tree Regression

Regression analysis aims to establish a relationship between a vector of independent variables (say $s$) and a vector of dependent variables (say $a$) by determining appropriate coefficients or parameters to model their association. We seek to minimize the sum of squared residuals between $a$ and $\hat{a}$, where $\hat{a}$ is the prediction of the model parameterized by $\theta$. Formally, we define the optimization problem as follows: $\boldsymbol{\theta}^* = \arg\min_{\boldsymbol{\theta}} \sum_{i=1}^{n} (a_i - \hat{a}_i(\mathbf{s}_i; \boldsymbol{\theta}))^2$, where, $\theta^*$ represents the optimal parameters that minimize the sum of the squared differences between the actual values $a_i$ and the predicted values $\hat{a}_i$ across all observations $i = 1, \ldots, n$.

A powerful approach to solving this regression problem is using regression trees, particularly through the XGBoost algorithm [19, 20, 21]. XGBoost, short for Extreme Gradient Boosting, constructs an ensemble of decision trees to model the relationship between $s$ and $a$. This method iteratively improves the model by minimizing the objective function using a gradient-based optimization approach. Mathematically, the model output $\hat{a}$ in XGBoost is expressed as the sum of predictions from K individual regression trees: $\hat{a}_i = \sum_{k=1}^{K} f_k(\mathbf{s}_i)$, where each $f_k$ represents a regression tree parameterized by its structure and leaf weights.

The objective function ($\mathcal{L}$) in XGBoost includes a regularization term to penalize model complexity, thereby preventing overfitting. The formal objective function to be minimized is:

$$\mathcal{L} = \sum_{i=1}^{n} (a_i - \hat{a}_i)^2 + \sum_{k=1}^{K} \Omega(f_k) \tag{1}$$

where $\Omega(f_k)$ is the regularization term for the k-th tree, dependent on number of leaves and the weight of the leaves. So each tree $f_k$ is optimized to correct the errors of the previous trees, resulting in a highly accurate and robust regression model. Furthermore, XGBoost incorporates techniques such as sparsity-aware algorithms for sparse data and weighted quantile sketch for approximate tree learning. It optimizes cache access, data compression, and sharding to build a scalable tree-boosting system, making it faster and more scalable than traditional boosting methods, even for large datasets with limited resources [21, 22].

## 2.3 Related Works

Several approaches have been explored for explainable policy learning, often using decision tree-based methods [23, 24, 25, 26]. Differentiable decision trees (DDTs) is a framework that integrates policy gradient [25], showing that such *online*-trained policies outperform MLP policies in both performance and confirmed usability and interpretability by a user-study with Likert scale ratings. Similarly, [27] examined neural policies' interpretability through disentanglement, extracting abstractions via decision trees based on neuron responses to provide clearer insights into the learned policies of robots.

**RWDTP:** Advantage-weighted regression (AWR) is an actor-critic algorithm that trains through supervised regression, showing strong performance in both online and offline settings [5]. AWR involves two supervised learning steps: regressing onto target values for the value function and

regressing onto weighted target actions for the policy. Initially introduced as an off-policy online algorithm, AWR uses advantage-weighted updates to improve the policy. TD3+BC combines the Twin Delayed Deep Deterministic Policy Gradients (TD3 [28]) algorithm with behavior cloning in the offline setting [29]. It adds a simple regression loss to the TD3 loss, regularizing the policy by cloning the behavior observed in the dataset.

Some prevalent approaches focus on reprioritizing the sampling of offline data to improve training. Return-based Data Resample (ReD) adjusts the probability of sampling each transition in the dataset according to its episodic return [30]. BAIL learns a V-function to select high-performing actions, which are then used to train the policy network using imitation learning [31]. Similarly, Offline Prioritized Experience Replay (OPER) employs priority functions to prioritize highly-rewarding transitions, ensuring they are more frequently visited during training [32, 33].

**RCDTP:** Upside-down reinforcement learning redefines cumulative rewards as inputs rather than predictions and adopts a 'true' supervised learning objective, ensuring stable and consistent learning targets [3]. This marks a shift in reinforcement learning methodologies by transforming how rewards and learning objectives are utilized [34]. An important work in this line, Decision Transformer (DT [4]), frames offline RL as a return-conditioned sequence modeling problem [35]. This eliminates the need for bootstrapping for long-term credit assignment and avoids discounting future rewards, preventing short-sighted decisions. Janner et al. [36] utilize a diffusion model (DDPM [37]) for trajectory generation, offering flexible behavior synthesis and variable-length planning. The denoising process allows flexible conditioning: either by using gradients of an objective function to bias plans toward high-reward regions or by conditioning on a specified goal. The guidance of the return-conditioning model is then injected into the reverse sampling stage. Despite being significant milestones in using Generative Models in RL, DT and Diffuser results often do not surpass 'conventional' methods like CQL and behavior cloning, even with larger models and increased training and inference times.

## 3  RWDTP and RCDTP frameworks

Training an off-policy reinforcement learning (RL) algorithm typically involves training a critic network to approximate the value function while simultaneously training an actor network to maximize this value. However, due to the lack of fresh interactions with the environment, the actor can learn *out-of-distribution (OOD)* actions that falsely appear to maximize the approximate value function due to estimation errors related to these actions. Contemporary offline RL algorithms address this overestimation problem through several strategies: regularizing the value function [16, 38, 39], constraining the action distribution [29, 40, 41], or avoiding the querying of out-of-distribution actions while training [4, 42, 43]. The decision tree policies (DTPs) trained in this study adopt the third strategy; as they do not rely on a value function, no OOD actions are queried while training them.

**RWDTP:** The policy is deterministic and is conditioned on the state, represented as $\hat{a}_t = \pi(s_t; \theta)$. The corresponding regression objective is:

$$J_{RWDTP}(\pi) = \sum_{n=1}^{N} (a_n - \pi(s_n; \theta))^2 \tilde{R}_n^p \tag{2}$$

where, $N$ is the number of datapoints in the dataset, $\tilde{R}_n \in [0, 1]$ is the discounted sum of future rewards normalized over the dataset and $p$ is a hyper-parameter that adjusts the distribution of $\tilde{R}_n$ as an exponent. Mathematically, for a reward of $r_t$ at timestep $t$ in a trajectory of length $T$:

$$\tilde{R}_t = \frac{R_t - \min_N\{R_n\}}{\max_N\{R_n\} - \min_N\{R_n\}} \text{ and } R_t = \sum_{k=t}^{T} \gamma^{k-t} r_k$$

**RCDTP:** In RCDTP, the action is conditioned on the return-to-go, and timestep in addition to the state. Return to go (RTG) for a timestep is the undiscounted sum of future rewards from that

timestep (i.e $R_t = \sum_t^T r_t$). The policy is represented as $\hat{a}_t = \pi(s_t, R_t, t; \theta)$. The corresponding regression objective is:

$$J_{RCDTP}(\pi) = \sum_{n=1}^{N} (a_n - \pi(s_n, R_n, t_n; \theta))^2 \qquad (3)$$

**Optimal Policy:** For both RWDTP and RCDTP, the optimal policy is obtained by solving the unconstrained minimization problem:

$$\pi_\theta^*(\cdot) = \text{argmin}_{\pi_\theta} J(\pi_\theta) \qquad (4)$$

A neural network aimed at minimizing the cost $J$ as described in equations 2 and 3 typically employs gradient descent or a similar method, updating the model parameters $\theta$ through the iterative rule $\theta \leftarrow \theta - \alpha \nabla_\theta J$.

However, when using decision trees, the gradients and Hessians need to be calculated with respect to the model output ($\hat{a}$) at every boosting round [21], rather than the model parameter $\theta$. These gradients and Hessians are then utilized in constructing additional trees until convergence. The output of the decision tree associated with the $k^{\text{th}}$ boosting round, with observations at step $i$, is given as $\hat{a}_i^k := \pi^k(s_i, R_i, i)$. The final policy will consist of a sum of such outputs,

$$\pi(\cdot) = \sum_{k=1}^{K} \pi^k(\cdot), \qquad (5)$$

for some fixed training budget, $K$, where each policy $\pi^k$ is referred to as a weak policy and is optimized according to the second-order approximation in equation 6 and using standard techniques for decision-tree splitting and optimization [21].

$$\pi^k = \text{argmin}_\pi \sum_{i=1}^{N} \left[ \nabla_{\hat{a}^{k-1}} J \cdot \pi(\cdot) + \frac{1}{2} \nabla_{\hat{a}^{k-1}}^2 J \cdot \pi(\cdot)^2 \right]. \qquad (6)$$

**Policy Training Implementation:** In this study, we use the Extreme Gradient Boosting (XG-BOOST) algorithm to fit the decision tree policies. RWDTP has four training hyperparameters: the discount factor ($\gamma$), the power applied to the normalized return, the number of weak policies/trees to be constructed, and the maximum depth of these trees. With the discount factor and return-power both set to 1, RCDTP has only two training hyperparameters. Additionally, RCDTP employs a runtime hyperparameter set at the beginning of the episode, the target return, which conditions the policy. This target return is the total return the agent is prompted to collect within the episode and is updated at each step by subtracting the reward obtained in the previous step ($R_{t+1}^{\text{Target}} = R_t^{\text{Target}} - r_t$).

## 4 Experimental Results

### 4.1 Gym Mujoco Locomotion

This section focuses on the results in Gym-Mujoco environments with the D4RL datasets [44], specifically Walker2d-v2, Hopper-v2, and Halfcheetah-v2. We include comparisons across four kinds of datasets commonly used for benchmarking: medium, medium replay, medium-expert, and expert. Medium and expert datasets are generated from rollouts of 'medium' and 'expert' performance policies, respectively, trained with Soft Actor-Critic. Medium-expert datasets are created by combining medium datasets with expert datasets, while medium-replay datasets consist of replay buffers generated during the training of medium policies.

Table 1 presents a comparison involving recent baselines EDAC and SAC-N, which exhibit superior performance in these tasks, alongside the conventional and widely used baseline, CQL [45, 16]. The mean and standard deviations are calculated from evaluations across five seeds. The training is performed on hardware comprising an Intel Core i9-13900KF CPU and an NVIDIA RTX 4090 GPU

Table 1: Results on Gym-MuJoCo Locomotion Tasks (All experiments are run by the authors.)

| Environment | Dataset | RWDTP | RCDTP | EDAC | SAC-N | CQL |
|---|---|---|---|---|---|---|
| HalfCheetah | Medium | $42.11 \pm 1.07$ | $41.44 \pm 0.72$ | 67.85 | **69.04** | 31.72 |
| | Medium-Replay | $40.03 \pm 1.75$ | $40.08 \pm 0.43$ | **63.52** | 63.39 | 44.81 |
| | Medium-Expert | $\mathbf{85.40 \pm 4.64}$ | $86.22 \pm 3.85$ | 49.73 | 61.53 | 27.20 |
| | Expert | $90.70 \pm 1.74$ | $91.24 \pm 0.45$ | 2.02 | 1.67 | **96.65** |
| Hopper | Medium | $55.01 \pm 3.51$ | $61.52 \pm 7.5$ | **102.39** | 9.68 | 77.42 |
| | Medium-Replay | $84.10 \pm 9.63$ | $83.88 \pm 7.25$ | 98.20 | **101.02** | 91.70 |
| | Medium-Expert | $\mathbf{111.24 \pm 0.84}$ | $111.13 \pm 1.63$ | 32.55 | 39.59 | 27.29 |
| | Expert | $\mathbf{111.39 \pm 1.29}$ | $110.92 \pm 0.76$ | 48.62 | 1.27 | 103.29 |
| Walker2d | Medium | $79.48 \pm 8.80$ | $79.62 \pm 7.2$ | **90.91** | 88.01 | 60.13 |
| | Medium-Replay | $60.96 \pm 22.81$ | $75.02 \pm 8.49$ | 80.55 | 81.12 | **82.32** |
| | Medium-Expert | $109.16 \pm 0.37$ | $85.99 \pm 8.522$ | 106.69 | **116.18** | 44.36 |
| | Expert | $\mathbf{108.60 \pm 0.75}$ | $108.15 \pm 0.38$ | 20.39 | 1.02 | 107.58 |
| | Average Training Time | $20.79 \pm 14.44$ s. | $104.14 \pm 115.90$ s. | $\geq$ 1hr. | $\geq$ 1hr. | $\geq$ 1hr. |

(24 GB), with computational resources for baselines restricted to one hour on the GPU to ensure a fair comparison. Two primary reasons motivate this approach: first, SAC-N, with a large number of critics and sufficient gradient steps, can outperform all other methods in locomotion tasks, necessitating an equitable comparison; second, our method emphasizes time-efficient training, thus necessitating performance assessment in a resource-limited setting. Interestingly, baseline performance sometimes improved under these constraints compared to the results reported in [46], where complete training results and additional baseline data are also available. RWDTP and RCDTP were trained on a CPU, completing training in just a few minutes, and demonstrated excellent performance, particularly in medium-expert and expert datasets. More experiments on enhancing DTP performance in the non-expert regime are discussed in Appendix A.7.

## 4.2 Gym Robot Manipulation - Adroit and Kitchen Tasks

The Adroit domain involves controlling a 24-DoF robotic hand, and includes four tasks: pen, door, hammer, and relocate. Each task presents unique challenges that require precise robotic manipulation and control. This section includes comparative results from PLAS [47], an offline RL algorithm excelling in manipulation tasks, alongside other baselines [38, 41] employed by PLAS.

Table 2: Results on Adroit Manipulation Tasks (Baseline results are from [17, 46, 47]).

| Environment | Dataset | RWDTP | RCDTP | EDAC | SAC-N | CQL | BCQ | PLAS |
|---|---|---|---|---|---|---|---|---|
| Pen | Expert | $\mathbf{120.65 \pm 32.81}$ | $112.21 \pm 25.04$ | -1.55 | 87.11 | -1.41 | 114.9 | **120.7** |
| Door | Expert | $105.00 \pm 0.67$ | $\mathbf{106.42 \pm 0.25}$ | 106.29 | -0.33 | -0.32 | 99.0 | 104.2 |
| Hammer | Expert | $\mathbf{126.00 \pm 2.28}$ | $125.23 \pm 1.02$ | 28.52 | 28.13 | 0.26 | 107.2 | **127.1** |
| Relocate | Expert | $\mathbf{110.62 \pm 1.83}$ | $109.72 \pm 2.19$ | 71.94 | -0.36 | -0.30 | 41.6 | 106.9 |

Franka Kitchen is a multitask environment featuring a 9-DoF Franka robot placed in a kitchen with some common household items. This is a sparse-reward manipulation setup with limited baselines for comparison across its three D4RL datasets: complete, partial, and mixed.

Table 3: Results on Franka Kitchen (Baseline results are from [47]).

| Environment | Dataset | RWDTP | RCDTP | BEAR | BCQ | PLAS |
|---|---|---|---|---|---|---|
| Kitchen | Complete | $50.0 \pm 0.0$ | $\mathbf{60.0 \pm 12.25}$ | 0.0 | 8.1 | 34.8 |
| | Partial | $\mathbf{45.0 \pm 24.49}$ | $40.0 \pm 12.25$ | 13.1 | 18.9 | **43.9** |
| | Mixed | $\mathbf{50.0 \pm 0.0}$ | $45.0 \pm 24.5$ | 47.2 | 8.1 | 40.8 |

## 4.3 Racecar Gym and Pybullet Drones

Figure 2a compares RCDTP with the Decision Transformer (DT [4]) in the F1tenth racing scenario where the total rewards is measured based on the percentage progress made on the specific racetrack/environment [48, 49]. Both return-conditioned methods are trained exclusively on a dataset from the Austria environment and their performance is tested in other racetracks. They rely solely on lidar and velocity observation without prior track-specific knowledge. RCDTP outperforms the Decision Transformer in this seen-to-unseen transferability test.

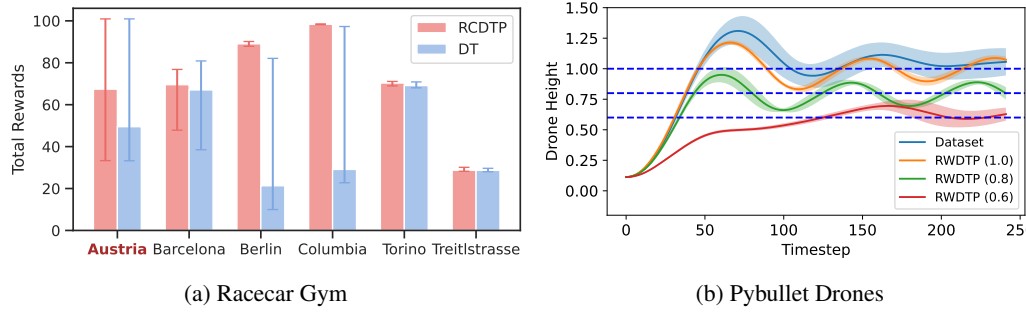

| (a) Racecar Gym | (b) Pybullet Drones |

Figure 2: Decision Tree Policies Applied to Wheeled and Flying Robots for Assessment of Zero-Shot Transfer of the Learned Policy. (a) Return-conditioning in Different F1tenth Racetracks. (b) Goal-conditioning for Different Heights Using RWDTP in the Pybullet Drones Simulation.

Figure 2b demonstrates the Pybullet drones simulation, where the objective is to hover at a certain height [50]. Although the observation space is 27-dimensional, only the height (z dimension) is shown for clarity. With a suboptimal demonstration dataset for hovering at 1m, we investigate whether the same dataset can train a decision tree policy to make the drone hover at a different height. By slightly modifying the observation space of RWDTP to incorporate the goal height, the policy can successfully transfer to hover at different heights without additional data collection, showcasing the generalization ability of decision tree policies.

## 5 Discussions

### 5.1 How do decision tree policies perform in delayed reward scenario?

In delayed reward scenarios where the agent receives 0 reward for all timesteps except the last one when it receives the cumulative reward, excelling requires strong long-term planning and effective use of past experiences. Unlike Q-learning variants, Behavior Cloning (BC) is agnostic to this delayed-reward transformation, and our experimental results in 4 show that RWDTP also remains nearly invariant. DTP-BC represents behavior cloning with decision trees and BC results are from the work of Tarasov et al. [17].

Table 4: Performance drop on delayed rewards scenarios

| Dataset | Hopper-M | | Hopper-MR | | Hopper-ME | | Hopper-E | |
|---|---|---|---|---|---|---|---|---|
| RWDTP | $53.96 \pm 6.01$ | ↓2% | $78.89 \pm 20.58$ | ↓6% | $110.37 \pm 0.64$ | ↓1% | $110.67 \pm 0.29$ | ↓1% |
| RCDTP | $49.78 \pm 1.21$ | ↓19% | $52.91 \pm 10.68$ | ↓37% | $46.65 \pm 6.77$ | ↓58% | $111.92 \pm 0.82$ | ↑1% |
| 100% BC | $53.51 \pm 1.76$ | 0% | $29.81 \pm 2.07$ | 0% | $52.30 \pm 4.01$ | 0% | $110.85 \pm 1.02$ | 0% |
| 100% DTP-BC | $49.33 \pm 4.04$ | 0% | $23.08 \pm 8.46$ | 0% | $53.26 \pm 6.04$ | 0% | $110.68 \pm 0.784$ | 0% |

### 5.2 Are decision tree policies only as effective as BC?

In the maze2d environment, a sparse-reward gym robotics environment, both RCDTP and RWDTP significantly outperform the Behavior Cloning (BC) variations, including Decision Transformer (DT) and behavior cloning with decision trees (DTP-BC), demonstrating superior long-term planning and utilization of past experiences (table 5).

Table 5: Results on Maze2d Against %BC

| Environment | Dataset | RWDTP | RCDTP | 100% DTP-BC | 100% BC | 50% BC | 10% BC | DT |
|---|---|---|---|---|---|---|---|---|
| Maze2d | Umaze | $64.74 \pm 6.31$ | $\mathbf{101.61 \pm 30.91}$ | $9.38 \pm 16.45$ | $0.36 \pm 8.69$ | $4.02 \pm 17.39$ | $12.18 \pm 4.29$ | $18.08 \pm 25.42$ |
| | Medium | $56.51 \pm 15.8$ | $\mathbf{63.37 \pm 56.02}$ | $11.61 \pm 8.77$ | $0.79 \pm 3.25$ | $11.15 \pm 8.06$ | $14.25 \pm 2.33$ | $31.71 \pm 26.33$ |
| | Large | $\mathbf{76.96 \pm 20.57}$ | $73.66 \pm 26.86$ | $3.85 \pm 8.94$ | $2.26 \pm 4.39$ | $-4.97 \pm 0$ | $11.32 \pm 5.10$ | $35.66 \pm 28.20$ |

### 5.3 How fast are decision tree policies in comparision to Sequence Modeling counterparts?

Table 6 presents a comparative analysis of our regression-based methods versus sequence modeling approaches, specifically Decision Transformer (DT [4]) and Trajectory Transformer (TT [43]). This comparison in Hopper expert datasets evaluates performance on the same seed, focusing on training time, inference time, and normalized returns. To match the performance of the regression-based

models, DT was trained for 30k gradient steps and TT for 10 epochs. Our key findings show that the CPU training time for RWDTP and RCDTP is less than $1\%$ of the GPU training time for DT and TT. In average, the *training time* for decision tree polices are less than the *inference time* for TT on the same CPU device. Appendix A.6 includes similar comparisons to other baselines.

Table 6: Training and Inference Time Comparison Against DT and TT

| Dataset | Hopper Expert | | | |
|---|---|---|---|---|
| Method | RWDTP | RCDTP | DT | TT |
| Training Device | CPU | CPU | GPU | GPU |
| Training Time $(\mu)$ | 9.64s | 11.36s | 2896.67s | 1961.87s |
| $(\sigma)$ | 1.94s | 2.09s | 34.29s | 1.34s |
| Inference Device | CPU | CPU | CPU | CPU |
| Inference Time $(\mu)$ | 3.15e-4s | 2.57e-4s | 2.30e-2s | 12.83s |
| $(\sigma)$ | 1.29e-3s | 5.63e-4s | 3.63e-2s | 1.11s |
| Normalized Returns | 111.81 | 113.68 | 116.25 | 110.82 |

## 5.4 Do RCDTP and RWDTP learn to prioritize similar features and predict similar actions?

Figure 3a shows the distribution of the action distance ($||a_A - a_B||_2$) evaluated over the observations in the Hopper-Expert dataset, where $A$ and $B$ are two different policies/sources. Our empirical experiments indicate that RCDTP and RWDTP may learn different policies depending on the dataset type, as they are conditioned on different inputs despite using similar regression objectives. However, the features that they learn to prioritize are fairly consistent. Feature-importance analysis in Hopper observation space reveals that both models prioritize the angular velocity and the angle of the foot joint the most when predicting the actions (figure 3b). These features closely correspond to the torque applied to the foot rotor in the action space.

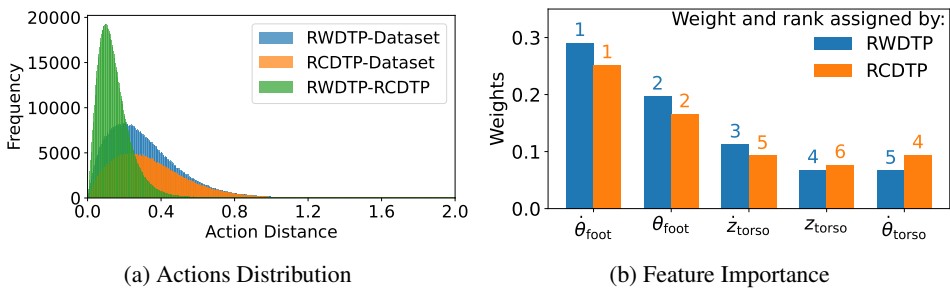

(a) Actions Distribution      (b) Feature Importance

Figure 3: Comparison Between Decision Tree Policies in Hopper Expert Dataset

## 5.5 Limitations

While the rapid training and inference capabilities make decision tree policies particularly suitable for quick experimentation in robot learning and real-time control of dynamical systems, there are some notable limitations. Decision Tree Policies are primarily applied to flat-structured observation spaces, and due to the choice of function approximator, our methods may not extend to more complex data modalities such as images and text. Future work could explore integrating these data types to enhance robot perception and decision-making.

Additionally, human behavior is often multimodal, with diverse actions in similar contexts, which our models may not fully capture as generative models do. Handling these multimodal datasets through energy-based methods remains a viable future direction. Furthermore, our training process is conducted offline, relying on pre-collected datasets. Due to the inherent characteristics of decision trees, which lack adjustable weights like those in neural networks, online training involving the construction of additional trees in each round appears impractical.

Despite these limitations, we believe that the proposed methods represent a significant advancement in integrating regression-based techniques into explainable offline RL, offering a robust foundation for future research and application in robot learning and control.

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

# A  Appendix

## A.1   XGBoost algorithm for state-action regression

Let $\mathcal{D} = <\mathbf{s}_i, a_i>_{i=1}^N$ denote the training dataset, where $\mathbf{s}_i$ represents the feature vector of instance $i$ and $a_i$ denotes its corresponding true target value. In XGBoost, the model output $\hat{a}$ is determined by aggregating predictions from K individual regression trees, denoted as $\hat{a}_i = \sum_{k=1}^K f_k(\mathbf{s}_i)$, where each $f_k$ represents a regression tree parametrized by its structure and leaf weights. Any tree $f_k$ is optimized to correct the errors of the previous trees, and is also referred to as a weak policy in this paper. To curb overfitting, XGBoost's objective function ($\mathcal{L}$) integrates a regularization term aimed at penalizing model complexity.

$$\mathcal{L}(\boldsymbol{\theta}) = \sum_{i=1}^n (a_i - \hat{a}_i)^2 + \sum_{k=1}^K \Omega(f_k) \tag{7}$$

where $\boldsymbol{\theta}$ represents the parameters of the model, $a_i$ is the true target value of instance $i$, and $\hat{a}_i$ is the predicted value. The regularization term $\Omega(f_k)$ for the $k$-th tree is typically defined as:

$$\Omega(f_k) = \gamma T_k + \frac{1}{2}\lambda \sum_{j=1}^{T_k} w_{kj}^2 \tag{8}$$

Here, $T_k$ denotes the number of leaves in the $k$-th tree, $w_{kj}$ represents the weight of the $j$-th leaf, $\gamma$ is a parameter controlling model complexity by penalizing the number of leaves, and $\lambda$ controls L2 regularization on leaf weights.

The optimization process in XGBoost proceeds by sequentially adding trees $f_k$ to fit the residual value that best reduce the objective function. The t-th tree is built to minimize the following objective:

$$\mathcal{L}^{(t)} = \sum_{i=1}^n \left[ g_i f_t(\mathbf{s}_i) + \frac{1}{2} h_i f_t(\mathbf{s}_i)^2 \right] + \Omega(f_t) \tag{9}$$

where $g_i$ and $h_i$ are the first and second-order gradients of the loss function with respect to the prediction at instance i.

$$g_i = \frac{\partial \mathcal{L}}{\partial \hat{a}_i^{(t-1)}}, \quad h_i = \frac{\partial^2 \mathcal{L}}{\partial \hat{a}_i^{(t-1)^2}} \tag{10}$$

where, $\hat{a}_i^{(t-1)}$ represents the predicted value of instance $i$ using the ensemble of $t-1$ trees.

## A.2   Policy Training

Decision tree policy training involves iteratively improving the ensemble model through a series of gradient-boosting steps and is detailed in algorithm 1. At each iteration, the algorithm computes the first and second-order gradients (gradients and hessians) of the loss function with respect to the current model's predictions. These gradients and hessians guide the construction of a new decision tree (a weak policy) that fits the residual errors of the current model. This new tree is then added to the ensemble, progressively refining the model's prediction. This iterative process continues until the specified number of estimators, $K$, is reached, thereby optimizing the model by sequentially minimizing the objective function.

Decision Tree Policies are generally not very sensitive to hyperparameters. Additionally, the training process is swift, making the tuning process easy and efficient if needed. For Gym-Mujoco locomotion tasks, RWDTP typically uses 100 estimators with a tree depth of 11, while RCDTP employs 1000 estimators and a tree depth of 6. We have found these values to be effective starting points for a range of tasks and often adjust the number of estimators based on dataset size. For example, adroit tasks, which involve smaller datasets, only require 50 estimators for RWDTP. Nonetheless, the configuration of 100 estimators and a tree depth of 11 for RWDTP, and 1000 estimators with a tree depth of 6 for RCDTP, offer robust baseline settings for hyperparameter tuning across most tasks.

**Algorithm 1:** Decision Tree Policy Training

---

**Input** : States **s**, Actions **a**, Target Returns **R**, Timesteps **t** (for RCDTP)

**Input** : States **s**, Actions **a**, Powered Return Weights $\tilde{\mathbf{R}}^{\mathbf{p}}$ (for RWDTP)

**Output:** Trained Decision Tree Policy

**Tree Hyperparameters:** Number of estimators **K**, Maximum Tree Depth **D**

**1. Preprocess Data**
   **For RCDTP:** $X\_train \leftarrow$ concatenate(**s**, **R**, **t**), $y\_train \leftarrow$ **a**
   **For RWDTP:** $X\_train \leftarrow$ **s**, $y\_train \leftarrow$ **a**

**2. Initialize Model**

   **For RWDTP:** $objective \leftarrow$ squared error weighted by $\tilde{\mathbf{R}}^{\mathbf{p}}$
   **For RCDTP:** $objective \leftarrow$ squared error
   $model \leftarrow$ Regressor($objective$, **K**, **D**)

**3. Train Model**
   **Function** `fit`(*model, X_train, y_train*)**:**
       **for** $t \leftarrow 1$ **to** $N$ **do**
           1. Compute gradients $g_i$ and hessians $h_i$
           2. Construct a new decision tree to fit the residuals
           3. Add the new tree to the model ensemble
       **end**
       **return** model
   $model \leftarrow$ `fit`($model, X\_train, y\_train$)

---

In addition to the hyperparameters related to estimators and tree depth, RWDTP incorporates an additional hyperparameter, $p$, which plays a role in weighting the samples based on their normalized return. Specifically, the dataset-normalized return is used with an exponential function of $p$ as a weight ($w = \tilde{R}^p$), providing a measure of how valuable each sample is relative to others in the dataset. Figure 4 illustrates the impact of this weighting mechanism, controlled by $p$ on normalized returns for different Walker2d datasets. The hyperparameter $p$ allows for adjusting this weighting, where higher values of $p$ amplify the prioritization of higher-return samples, effectively skewing the contribution of these samples during training. This enables the model to focus more on higher-return demonstrations. When $p$ is set to 0, the RWDTP method reduces to pure behavior cloning, where all samples contribute equally, regardless of their return.

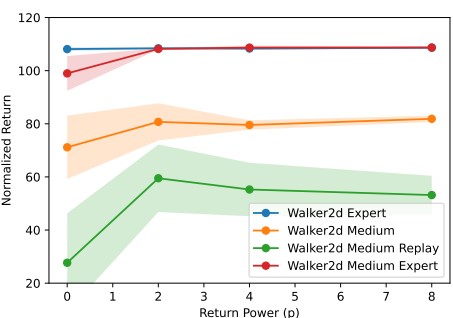

Figure 4: Impact of Hyperparameter $pp$ on Normalized Returns During Evaluation

### A.3 RCDTP: Modeling of Returns Distribution

To assess RCDTP's efficacy in modeling returns distributions, we generate plots illustrating the accumulated actual returns achieved by the agent when conditioned on specific target RTG values. For comparison, we also include the Decision Transformer (DT) return distribution plot for the same dataset. Although Decision Transformer uses extended context length, uses a larger model, and takes

50 times more training time, by visual comparison, it is evident that RCDTP's performance is closer to the desired return values, showing a comparable capacity for modeling of returns.

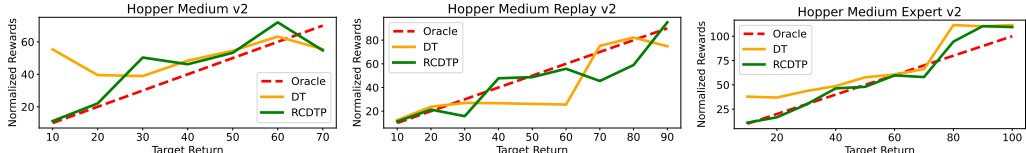

Figure 5: Returns Distributions Comparison Between RCDTP and Decision Transformer

## A.4 Explainability of Policies

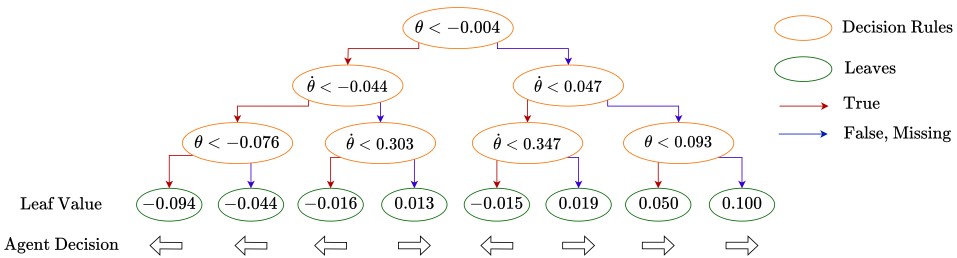

Figure 6: A Weak Policy Example in Cartpole Environment

Decision trees inherently offer greater explainability for policies. Figure 6 illustrates a decision tree from an ensemble used to form an expert-level policy in the CartPole environment. This specific tree shows that the agent's decisions are primarily based on the pole's angle: if the angle is negative, the agent is likely to push the cart left, and if positive, it mostly pushes the cart right. This results in an intuitive and interpretable weak policy. As more trees are added to the ensemble, the policy's strength and robustness increase. Feature importance analysis in figure 7 further enhances explainability, revealing that pole angle and angular velocity are more critical for stabilizing the pole than cart position and velocity. Similarly, in the pendulum environment, the policy equally prioritizes the cosine and sine components of the pendulum angle, giving substantial weight to all dimensions of the observation space in decision making.

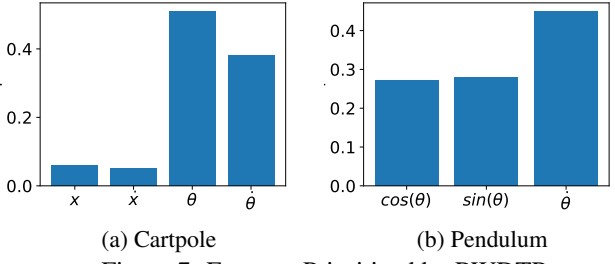

(a) Cartpole  (b) Pendulum

Figure 7: Features Prioritized by RWDTP

We have also visualized the decision boundaries learned by the decision tree policies. For CartPole, the decision boundary (whether to push the cart to the right) is visualized in figure 8a based on different pole angles and velocities, while keeping the cart's position and velocity fixed at (0,0). For the pendulum environment, we plot, in figure 8b, the predicted torque values over various pendulum angles and angular velocities, providing further insight into the model's action prediction surfaces. These visualizations further demonstrate the explainability of our approach across environments, highlighting how decision tree policies make control decisions.

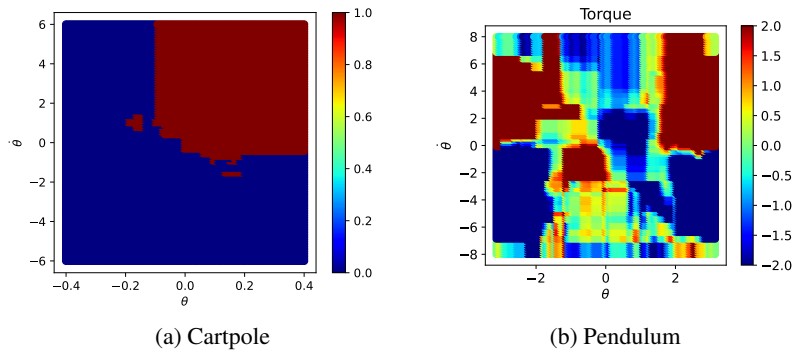

(a) Cartpole         (b) Pendulum

Figure 8: Decision Boundaries/Surfaces Learnt by RWDTP

## A.5   Racecar Gym and Pybullet Drones Experiments Details

The offline dataset for the Racecar Gym experiments was obtained by implementing a controller tailored to the Austria racetrack. This controller receives the raceline to be tracked and determines the steering angle and motor command based on the current heading angle and lookahead points in the raceline. After training offline on a hundred episode demonstrations collected using this controller in the Austria racetrack, the agents undergo a zero-shot transfer test on various other racetracks. Both RCDTP and DT agents utilize only Lidar and velocity observations, excluding raceline or map information, to predict actions, which include steering angle and motor command. In this environment, agents are rewarded based on racetrack progress and are penalized for collisions, leading to episode termination.

We conducted experiments using the hover-aviary environment in Pybullet drones. The target position is set to $[0, 0, 1]$, with the reward based on the L4 norm distance between the drone positions and the target. The primary objective is to maintain a hovering position around $z_{target} = 1$, with minimal deviations in the x and y dimensions from the starting state. Initially, we trained Proximal Policy Optimization (PPO [51]) for 100,000 timesteps to obtain a 'suboptimal' agent and collected 100 episodes to form an offline demonstration dataset. Subsequently, we modified the z dimension of the dataset using the transformation $z_{target} - z$ so as to incorporate the goal height ($z_{target} = 1$). After training RWDTP on this dataset with slightly modified observation space, we conducted simulations with $z_{target}$ values of 1, 0.8, and 0.6. The same RWDTP policy hovered the drone at each target height without any policy retraining, modifications to the reward function, or additional dataset collection.

## A.6   Training and Inference Time

Table 7 presents the training times (on GPU) and inference times (on CPU) for 1 million gradient steps across various methods, as recorded in our experiments. For baselines, we compare CQL, Behavioral Cloning (BC), and different variants of EDAC with 10 and 50 critic networks. The experiments were conducted on a system running Ubuntu 22.04.3 LTS, equipped with a 13th Gen Intel Core i9-13900KF processor with 24 cores, an NVIDIA RTX 4090 GPU (24 GB), and 64 GB of DDR5 memory.

Table 7: Training and Inference Time (in seconds)

| Method | Training Mean | Training Std | Inference Mean | Inference Std |
|--------|---------------|--------------|----------------|---------------|
| CQL | 6053.3 | 28.3 | 3.6e-4 | 1.3e-4 |
| BC | 692.26 | 13.45 | 1.04e-4 | 3.32e-5 |
| EDAC-10 | 3754.5 | 39.8 | 4.2e-4 | 5.8e-4 |
| EDAC-50 | 5154.0 | 62.74 | | |

## A.7 Using IQL Critic to Guide Action Selection

In our attempt to improve the performance of decision tree policies (DTPs) on datasets without expert-level demonstrations, we conducted additional experiments using an IQL [42] critic to guide action selection. Unlike traditional methods, IQL allows training the Q function without requiring an actor to sample unseen actions, providing a distinct advantage in offline learning. Specifically, we integrated a perturbation layer with a mean of the predicted action and a standard deviation of 0.2 into the output of RCDTP and RWDTP methods. From these perturbed actions, we sampled $N = 10$ actions and selected the one with the highest Q value. This action selection (or policy extraction) method, based on sampling, has been employed before in [52, 53, 54]. The performance of these IQL-assisted DTPs is denoted by the suffix of $+IQL$ in the table 8. Although the performance of 'plain' decision tree policies is on par with the conventional baselines in the non-expert regime, results of our experiments sometimes show performance improvements of up to $60\%$ when using the IQL critic. This also highlights the better performance of decision tree policy prior in comparison to the diffusion BC prior used in IDQL [53].

Table 8: Improved Performance in Non-expert Regime (Baseline results are from [53])

| Dataset | RWDTP | RCDTP | RWDTP+IQL | RCDTP+IQL | IQL | IDQL | CQL | DT | BC |
|---|---|---|---|---|---|---|---|---|---|
| Halfcheetah Med | 42.11 | 41.44 | 49.38 | 48.83 | 47.4 | **51.0** | 44.0 | 42.6 | 43.1 |
| Hopper Med | 55.01 | 61.52 | 87.04 | **88.58** | 66.3 | 65.4 | 58.5 | 67.6 | 63.9 |
| Walker2d Med | 79.48 | 79.62 | **87.96** | 83.07 | 78.3 | 82.5 | 72.5 | 74.0 | 77.3 |
| Halfcheetah Med-Rep | 40.03 | 40.08 | 45.39 | 43.91 | 44.2 | **45.9** | 45.5 | 36.6 | 4.3 |
| Hopper Med-Rep | 84.10 | 83.88 | 89.59 | **96.27** | 94.7 | 92.1 | 95.0 | 82.7 | 27.6 |
| Walker2d Med-Rep | 60.96 | 75.02 | 78.56 | 79.53 | 73.9 | **85.1** | 77.2 | 66.6 | 36.9 |
| Average | 60.28 | 63.59 | **72.99** | **73.37** | 67.47 | 70.33 | 65.45 | 61.68 | 42.18 |

To understand the performance gap between decision tree policies and their +IQL counterparts, we examine the distribution of actions learned by RWDTP and RWDTP+IQL in Hopper-Medium, where the gap is high, and Hopper-Medium-Replay, where the gap is lower. Figure 9 depicts the distribution of action distance ($||a_{policy} - a_{dataset}||_2$) evaluated over the observations in the dataset. Specifically, we examine the Hopper Medium and Medium Replay scenarios to illustrate the impact of potential overfitting on the performace. RWDTP's distribution of action distances from the dataset is highly skewed towards zero for Hopper-Medium compared to Hopper-Medium-Replay, indicating a lack of out-of-distribution generalization and potential overfitting within the dataset's distribution. As a result, when the IQL critic is used to guide the action selection, it shifts this action-distance distribution away from zero and also greatly improves performance.

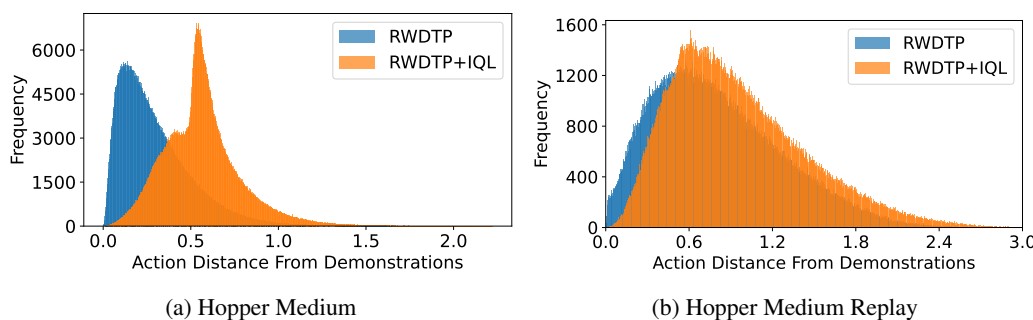

(a) Hopper Medium                 (b) Hopper Medium Replay

Figure 9: Distribution of Action Distance From Demonstrations

## A.8 Additional Feature Importance Analysis

Feature importance analysis can be a crucial tool in robot learning, particularly for understanding which input features are most influential in the decision-making process of learned policies. By identifying key features, we can better assess the interpretability and robustness of these policies, which is essential for ensuring interoperability across different tasks and environments. In this

additional study, we applied feature importance analysis to the Expert and Medium datasets in the Hopper and Walker2d environments to evaluate the behavior of the decision tree policies.

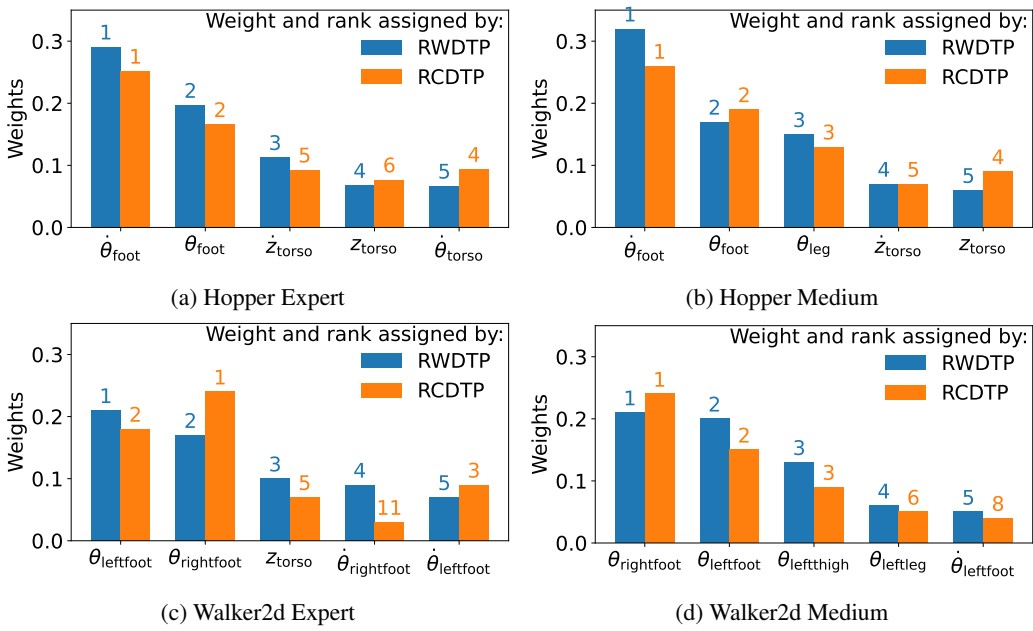

Figure 10: Feature Importance

Figures 10a through 10d display the ranked importance of key features identified by both methods. Our analysis reveals that despite being trained on different performance datasets, RWDTP and RCDTP tend to prioritize similar features when predicting actions. For each environment, the two most important features identified by both methods (DTPs) in both datasets remain the same. The rankings of important features as identified by the two policies in the Hopper datasets are summarized in table 9. This kind of consistency can be important for ensuring the interpretability of policies before deploying them to real-world robotic tasks.

Table 9: Feature Importance Rank Assigned in Hopper Datasets

| Observation | RWDTP (Expert) | RWDTP (Medium) | RCDTP (Expert) | RCDTP (Medium) |
|---|---|---|---|---|
| Angular velocity of the foot hinge | 1 | 1 | 1 | 1 |
| Angle of the foot joint | 2 | 2 | 2 | 2 |
| Velocity of the z-coordinate of the torso | 3 | 4 | 5 | 5 |
| Z-coordinate of the torso | 4 | 5 | 6 | 4 |

