# OpenReview forum: "Solving Offline Reinforcement Learning with Decision Tree Regression"
_robot-learning.org/CoRL/2024/Conference — CoRL 2024_

### Official Review · Reviewer_c4Yz · 2024-07-15
**good paper on a classical ML approach for offline RL datasets**

**Originality:** 3
**Technical Quality:** 3
**Clarity Of Presentation:** 4
**Potential Impact:** 3
**Recommendation:** 3
**Confidence:** 4

**Review:**

EDIT: Rebuttal acknowledged. I keep my accept recommendation as is.

In this paper, the authors train decision trees on low-dimensional offline RL tasks like the Mujoco locomotion tasks and Maze2d. They demonstrate surprisingly strong performance, often on-par with modern deep offline RL algorithms, especially on the datasets that are closer to expert-level. I think this paper is a good contribution to the conference. The audience will be interested in hearing about these results, and they call into question why we, as a community, are pouring so many resources into training deep models for low-dimensional offline RL. I have some concerns around strong claims, unclear assumptions, and the experiments section, which I highlight below.

Major Comments:
* Some strong claims are missing citations in the introduction. In particular, the lines "in many cases Behavior Cloning (BC) outperform offline RL algorithms" and "decision trees known for their superior performance in regression and classification tasks compared to neural networks".
* I think it would be good to discuss earlier on that decision trees are only applicable to low-dimensional offline RL tasks, i.e., tasks where the observation vectors are of a small dimensionality. A huge advantage of neural network function approximators is that they can learn directly from pixels as well, like in the original DQN paper. I know the authors discuss this in Section 5.5, but this feels too late to me; it's better to state the assumptions of your proposed approach earlier.
* A question I had throughout your experiments: Have you tried ignoring the reward signal and just doing pure behavior cloning with decision trees? I see that in Table 5, you are comparing RWDTP/RCDTP against BC, but I'm unclear if this is neural network BC or decision tree BC. To my understanding, there are two independent axes at play here: function approximator = {neural network, decision tree}; optimization strategy = {uses reward signal, pure behavior cloning}. Is this understanding correct?
* I would appreciate more discussion in the experiments section about _why_ RWDTP/RCDTP perform best when the data is more expert-level (stated in L198).
* Across the Experimental Results and Discussions sections, I felt that there is too wide a variety of domains, with some questions in the Discussions only being answered for 1 domain. In these cases, it would be better to provide a fuller set of results in the Experimental Results, and reference those in Discussions. For example, in Table 5 you only show BC results for Maze2d. Why was Maze2d chosen for this experiment? Why not include BC results for all domains in Table 1-3?
* Could you include a comparison to Decision Transformer for Table 1-3 as well?
* Eq 4: I got confused here. What happened to the $\tilde{R}_n^p$ from Eq 2? It seems like that got removed in Eq 4.
* Could you include some offline RL algorithms like CQL and EDAC to Table 6 (training and inference times) as well?
* Personally, I find the explainability aspect of decision trees to be the most exciting part of this work. Therefore, Fig 3 was very interesting to me. Would it be possible to also see this for a non-expert dataset, where decision trees didn't perform as well, like Hopper Medium? I wonder if we'd see the same qualitative results on that dataset.

Minor Comments:
* L26: "depending on the setting" It's more of a choice of which RL algorithm to use (which is decided by the researcher) rather than the problem setting (which is fixed & given to the researcher).
* You may be interested in checking out the paper "[When should we prefer Decision Transformers for Offline Reinforcement Learning?](https://arxiv.org/abs/2305.14550)"
* Eq 5: Shouldn't this be an average instead of a sum? If it's a sum, I don't understand why summing over actions suggested by each weak policy would make sense.
* L173: I'm unclear why RCDTP _needs_ $\gamma=1$. Can't we also do it with $\gamma<1$, by simply defining the return-to-go as a discounted sum?
* Figure 1: It would be better to show return on y-axis, and expert return as a dashed horizontal line
* L103: "the the" --> "the"
* L108: "XGBoost is faster" than what?
* L152: make sure to define $\gamma$ as the discount factor somewhere
* L152: "discounted sum of future returns" do you mean "rewards" instead of "returns"?
* Figure 2b: Wouldn't it be more natural to conduct this experiment with RCDTP instead of RWDTP? That way you can modulate by simply passing in a different target return. Then you don't have to "slightly modify the observation space" (L218).

**Quality Of The Limitations Section:**

3

**Questions For Rebuttal:**

See questions above, particulary in the Major Comments.

**Robotics Focus:**

3

**Summary Of Paper:**

Using gradient-boosted decision trees to solve offline RL tasks.

**Summary Of Recommendation:**

Surprisingly strong performance by decision trees, which could generate good discussion at the conference.

---

### Official Review · Reviewer_i7UK · 2024-07-19
**Interesting paper w/ opportunities for further investigation**

**Originality:** 3
**Technical Quality:** 3
**Clarity Of Presentation:** 3
**Potential Impact:** 3
**Recommendation:** 4
**Confidence:** 4

**Review:**

- The paper is well-structured and easy to read. I appreciate the “simplicity” of the approach, departing from the more standard offline RL algorithmic design
- The related work section should be extended to cover prior directions on using decision trees either directly for control or for post-hoc interpretability (e.g., [1]-[4])
- The baseline results on e.g. Gym-MuJoCo tasks are restricted to 1 hour training and often underperform the performance reported by the original authors (e.g. in Expert settings). It would be great to also report baselines results at convergence (w/ their associated run-time), which would increase transparency and allow others to make an informed decisions regarding trade-offs
- The F1 transfer experiments are very nice as they suggest the approach generalizing beyond the training distribution. It would be interesting to see how competitive the agent is, as it may learn only very simple heuristics. A trajectory overlay on a non-Austria track with a non-Austria expert policy and the RCDTP policy (+ video?) could be very informative.
- Furthermore, analyzing the underlying decision boundaries similar to the autonomous driving setting in [4] would be useful (similar to the Cartpole analysis in the Appendix, which should be prominently referenced in the main text).
- While generalization for racing and drone flight are promising, a robustness analysis on the MuJoCo and/or Adroit tasks would further strengthen the results (e.g., noisy sensor readings or system delays).
- The reduction in performance on the non-expert datasets (e.g. Table 1) could indicate that clean, uni-modal data is needed. The discussion here should be extended.
- It would be nice to include training times for the BC experiments


**References:**

[1] Z. Ding, et al. “CDT: Cascading decision trees for explainable reinforcement learning.” arXiv, 2020.

[2] A. Silva, et al. “Optimization methods for interpretable differentiable decision trees applied to reinforcement learning.” PLMR, 2020.

[3] A. Pace, et al. “Poetree: Interpretable policy learning with adaptive decision trees.” arXiv 2022.

[4] T.-H. Wang, et al. “Measuring interpretability of neural policies of robots with disentangled tree representations.” CoRL, 2023.

**Quality Of The Limitations Section:**

3

**Questions For Rebuttal:**

- I might have missed this, but over how many seeds/re-runs were the standard deviation results computed?
- In the drone scenario, how was the goal height incorporated into the observations (agent height and difference to goal height)? Which initial conditions (agent height) does the expert data contain?
- At which dimensionality of both the state and action space would the method reach its limits?
- What are the values for the number of estimators and tree depth for each task?
- In Algorithm 1, does N refer to K?

**Robotics Focus:**

3

**Summary Of Paper:**

This paper proposes methods to solve offline reinforcement learning problems via decision tree representations, incorporating return information either via conditioning (RCDTP) or by weighting the optimization objective (RWDTP) and running XG-BOOST. The methods are tested on a variety of continuous control offline RL task and display favorable performance, particularly in the expert demonstration regime, while significantly cutting down on training time.

**Summary Of Recommendation:**

I lean towards accept as the method’s simplicity is a nice departure from the more conventional offline RL setting, with very fast training times that results in favorable policy performance. Quantitative and qualitative analysis could be strengthened for full transparency and higher impact.

---

### Official Review · Reviewer_NHhZ · 2024-07-21
**Solving Offline Reinforcement Learning with Decision Tree Regression**

**Originality:** 3
**Technical Quality:** 3
**Clarity Of Presentation:** 3
**Potential Impact:** 3
**Recommendation:** 3
**Confidence:** 2

**Review:**

This paper proposes using Decision Tree Regression supervised learning to learn optimal and explainable policy using offline data.
The approach shows superior performance on D4RL datasets for locomotion and manipulation, as well as other robotic tasks involving wheeled and flying robots.

Strengths:
 - The paper is well-written and easy to follow.
 - The approach shows strong performance on D4RL datasets for a variety of robotic tasks.
 - The method provides explainable policies, which is beneficial for understanding decision-making.

Weaknesses
 - There are concerns about overfitting and underfitting when using offline data.
 - Questions remain about the effectiveness of the method when the offline data is not from expert trajectories.
 - The paper lacks clarity on the intuition behind using reward as the weight in the RWDTP method.
 - The approach's ability to avoid out-of-distribution actions is noted, but its limitations with non-expert data are uncertain.

Overall, the paper is well-written and easy to follow. The work is sound, and we can get an explainable policy using this method, but I am not sure about the overfitting and underfitting problems of offline data.

**Quality Of The Limitations Section:**

2

**Questions For Rebuttal:**

- In the RWDTP method, what is the intuition of using reward as the weighted for the lost?
 - The decision tree policies trained in this paper are based on avoiding out-of-distribution action distribution.
 - Do you have problems with overfitting and underfitting when learning the optimal policy? What if the offline data collected is not from the expert trajectory?  Is it true that our performance is not competitive when we do not have an expert dataset?

**Robotics Focus:**

2

**Summary Of Paper:**

This paper presents a method using Decision Tree Regression to learn optimal and explainable policies from offline data, showing strong performance on D4RL datasets, but raises concerns about potential overfitting and underfitting issues with non-expert data.

**Summary Of Recommendation:**

I would recommend weak reject and wait for the rebuttal from the author.

---

### Author Rebuttal · Authors · 2024-08-04

Dear Reviewers,

We would like to express our sincere gratitude to all of you for your thorough and insightful feedback on our manuscript. We are pleased that you found the paper to be an interesting read. In response to some common concerns raised across the reviews, we have provided the following clarifications and updates. For your convenience, a PDF version of our response is also available as the accompanying rebuttal document.


One recurring concern was the performance of decision tree policies on non-expert datasets within the gym-mujoco environments. A key challenge in offline RL is learning an optimal policy from a dataset with suboptimal demonstrations. The critic, typically trained via dynamic programming, usually helps the actor by guiding it and stitching suboptimal trajectories into a more optimal policy [1,2,3]. We hypothesize that our approach, which does not incorporate such a critic, might produce suboptimal policies when the dataset lacks the explicit presence of expert actions, even though it may still minimize the regression objective we established. As noted in [4], methods that constrain policies to the distribution of the dataset often suffer from limited exploration, affecting policy improvement. Even with a critic, Q-learning can converge towards a suboptimal policy due to the constraints imposed by the actor's exploration limitations [4].

To address this, we have been exploring methods to enhance the performance of decision tree policies in the absence of an expert dataset. Specifically, we conducted an experiment using a Q network (NN) trained via IQL [5] to guide action selection, inspired by the work of [6]. Unlike traditional approaches, IQL does not require an actor to sample (unseen) actions to train the approximate Q function. In our experiment, we incorporate a perturbation layer (mean=predicted action, std=0.2) to the output of our RCDTP and RWDTP agents, sampling N (=10) actions and selecting the one with the highest Q value. This process is somewhat analogous to IDQL, which employs a policy previously trained using diffusion BC. We add the suffix +IQL in the results below, where decision tree policies are coupled with IQL critic for action selection.

***Page (1/3)***

---

### Decision · Program_Chairs · 2024-09-04

**Decision:**

Accept

**Comment:**

All reviewers are excited about this work. It dares to go against the current flow of using neural networks for low-dimensional tasks, and demonstrates strong merits of using decision trees. It achieves superior performance and yields explainable policies. This excitement is somewhat mitigated by a long list of technical questions by all reviewers, including recurring questions around reliance on expert demonstrations.

The authors provide additional detail and clarifications in their rebuttal. Acceptance would be on faith that these will be integrated into the paper as promised.

Some particular strengths of this paper:
- well written and easy to follow,
- strong performance on diverse robotic tasks,
- the method provides explainable policies,
- fast training times.

Questions mostly concerned technical details and were largely answered convincingly by the authors.